# Visual-based Policy Learning with Latent Language Encoding

**Jielin Qiu\*** [1]  **Mengdi Xu\*** [1]  **William Han\*** [1]  **Bo Li** [2]  **Ding Zhao** [1]

## Abstract

Large Language models (LLMs) have shown remarkable success in assisting robot learning tasks, i.e., complex household planning. However, the performance of pretrained LLMs heavily relies on domain-specific templated text data, which may be infeasible in real-world robot learning tasks with image-based observations. Moreover, existing LLMs with text inputs lack the capability to evolve with non-expert interactions with environments. In this work, we introduce a novel learning paradigm that generates robots' executable actions in the form of text, derived solely from visual observations, using language-based summarization of these observations as the connecting bridge between both domains. Our proposed paradigm stands apart from previous works, which utilized either language instructions or a combination of language and visual data as inputs. Moreover, our method does not require oracle text summarization of the scene, eliminating the need for human involvement in the learning loop, which makes it more practical for real-world robot learning tasks. Our proposed paradigm consists of two modules: the SUM module, which interprets the environment using visual observations and produces a text summary of the scene, and the APM module, which generates executable action policies based on the natural language descriptions provided by the SUM module. We demonstrate that our proposed method can employ two fine-tuning strategies, including imitation learning and reinforcement learning approaches, to adapt to the target testing tasks effectively. We conducted extensive experiments involving various SUM/APM model selections, environments, and

tasks across 7 house layouts in the VirtualHome environment. Our experimental results demonstrate that our method surpasses existing baselines, confirming the effectiveness of this novel learning paradigm.

## 1. Introduction

There has been a surge of interest in building Large Language Models (LLMs) pretrained on large-scale datasets and exploring LLMs' capability in various downstream tasks. LLMs start from the Transformer model (Vaswani et al., 2017b) and are first developed to solve different natural language processing (NLP) applications (Devlin et al., 2019; Liu et al., 2019; Brown et al., 2020). Recently, LLMs also show great potential for accelerating learning in many other domains by generating learned embeddings as meaningful representations for downstream tasks and encoding transferable knowledge in large pretraining datasets. Examples include transferring the knowledge of LLM to, i.e., robotics control (Liang et al., 2022; Ahn et al., 2022), multimodal learning (Zeng et al., 2022; Zellers et al., 2021), decision-making (Li et al., 2022b; Huang et al., 2022a), code generation (Fried et al., 2022), laws (Kaplan et al., 2020), computer vision (CV) (Radford et al., 2021), and so on.

In this paper, we focus on the problem of facilitating robot learning by having a LLM in the loop. The robot generates actions according to its environment observations, which are, in general, sensory information in the format of images, point clouds, or kinematic states. We identify one key challenge in massively deploying LLMs to assist robots is that *LLMs lack the capability to understand such non-text-based environment observations*. To solve this challenge, Liang et al. (2022) utilize rule-based perception APIs to transform image-based observations into text formats, which then serve as inputs to the LLM. We instead propose to integrate the multimodal learning paradigm to transform images into texts, which allows more principled and efficient transfer to novel robot learning tasks than rule-based APIs. Another key challenge is *the widely-existing large distribution shifts between the training tasks of large pretrained models and testing tasks in the domain of robot learning*. To close the domain gap, Li et al. (2022b) adapt the pretrianed LLM to downstream tasks via finetuning with observations converted into text descriptions. In the presence of realis-

---

[*]Equal contribution  [1]Carnegie Mellon University [2]University of Illinois Urbana-Champaign. Correspondence to: Jielin Qiu, Mengdi Xu, William Han <jielinq,mengdixu,wjhan@andrew.cmu.edu>.

*Proceedings of the 39th International Conference on Machine Learning*, Baltimore, Maryland, USA, PMLR 162, 2023. Copyright 2023 by the author(s).
Interactive Learning with Implicit Human Feedback Workshop at ICML 2023.

tic visual observations, it is still being determined what is an appropriate method to co-adapt pretrained foundation models for testing tasks in robot learning.

To address the above challenges, we propose a new visual-based robot learning paradigm that takes advantage of embedded knowledge in both multimodal models and LLMs. To align different modalities in the visual observations and text-based actions, we consider language as the bridge information. We build a scene-understanding model (SUM) with a pretrained image captioning model to grant the robot the ability to describe the surrounding environment with natural language. We then build an action prediction model (APM) with a LLM to generate execution actions according to the scene caption in the format of natural language. To adapt per-tained models in SUM and APM to downstream robot learning tasks, we propose to finetune the multimodal model in SUM with pre-collected domain-specific image-caption pairs and the language model in APM with corresponding language-action pairs. Besides finetuning with expert demonstrations, we further propose a finetuning paradigm of APM based on the sparse environment feedbacks to endow APM's capability to evolute with non-expert data. An illustration of the proposed framework is Figure 1.

Our contributions are summarised as follows:

- We introduce a novel robot learning paradigm with LLM in the loop that handles multiple modalities of visual observations and text-based actions in a principled manner. We bridge both modalities with natural language generated by a pretrained multimodal model.

- To adapt to target testing tasks, we propose two fine-tuning strategies, including imitation learning and reinforcement learning approaches. We collect a new expert dataset for imitation learning-based finetuning.

- We test the adaptation performance of multiple models of SUM and APM in seven house layouts in the VirtualHome environment. Our experiments demonstrate that our proposed paradigm shows promising results.

## 2. Related Work

**Language Models in Robot Learning**   Recently, several works have successfully combined LLMs with robot learning by taking advantage of the knowledge learned by LLMs i.e., reasoning (Liang et al., 2022; Zeng et al., 2022; Zellers et al., 2021), planning (Shah et al., 2022; Huang et al., 2022b; Kant et al., 2022; Li et al., 2022b; Huang et al., 2022a), manipulation (Shafiullah et al., 2022; Jiang et al., 2022; Shridhar et al., 2022; Bucker et al., 2022; Ren et al., 2022; Tam et al., 2022; Khandelwal et al., 2022; Shridhar et al., 2021; Xu et al., 2022; 2023), and navigation (Lin et al., 2022; Parisi et al., 2022; Gadre et al., 2022; Hong et al., 2021; Majumdar et al., 2020), which demonstrated the feasibility of using LLM to assist robot learning.

**Visual Feedback in Robot Learning**   Visual feedback is commonly used in robot learning. Gothoskar et al. (2020) learned a generative model from actions to image observations of features to control a robot from visual feedback. Ma et al. (2022) proposed a self-supervised pretrained visual representation model which is capable of generating dense and smooth reward functions for unseen robotic tasks. Strokina et al. (2022) reviewed the methods of reward estimation and visual representations used in learning-based approaches for robotics applications. Mohtasib et al. (2021) studied the performance of dense, sparse, visually dense, and visually sparse rewards in deep RL.

**Pre-training and Fine-tuning of Language Models** Over the past few years, fine-tuning (Howard & Ruder, 2018) has superseded the use of feature extraction of pretrained embeddings (Peters et al., 2018) while pretrained language models are favored over models trained on many tasks due to their increased sample efficiency and performance (Ruder, 2021). The success of these methods has led to the development of even larger models (Devlin et al., 2019; Raffel et al., 2019). But those large models may not perform well on data that is different from what they were pretrained on. Under this case, fine-tuning pretrained contextual word embedding models to supervised downstream tasks has become commonplace (Hendrycks et al., 2020; Dodge et al., 2020). More related works can be found in Appendix B.

## 3. Method

In this section, we first introduce our focused problem in Section 3.1, which is generating a visual-based policy by leveraging pretrained large models. We then introduce SUM, which learns language descriptions of the surrounding environment in Section 3.2, and APM which predicts actions based on SUM's caption output in 3.3. To grant both SUM and APM the capability of making the correct understanding and decision in the target domain, we propose finetuning algorithms in Section 3.2 and 3.3. Our code and data are provided in the supplementary materials.

### 3.1. Problem Formulation

We consider a general and realistic robot learning task where a robot agent receives a sequential visual observation $V = [v_1, v_2, ..., v_t]$, where $t$ is the timestep, and aims to generate a sequence of actions $A = [a_1, a_2, ..., a_t]$ based on the pure visual observations $V$. Traditionally, the robot's policy is trained from scratch in the target tasks. Inspired by the success of large pretrained models, we aim to explore the benefit of utilizing pretrained LLMs and multimodal models for general robot learning tasks, where only visual observations are available as inputs. Given the prevailing domain shift between the training domain of the pretrained

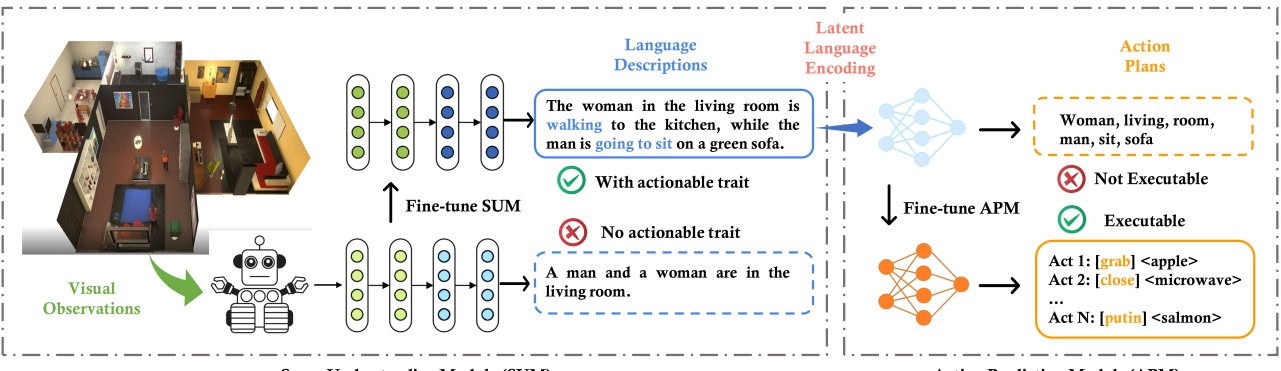

*Figure 1.* The overall architecture of our approach, which includes a scene understanding module (SUM) and an action prediction module (APM). The agent takes pure visual observations and encode the information as latent language, then the language is transferred to APM for action generation. APM fine-tuned on VirtualHome can generate executable action plans directly.

models and the robot learning tasks, we are motivated to develop a principled finetuning method.

### 3.2. SUM: Learning Scene Descriptions from Visual Observations into Language

The goal of the SUM (scene understanding module) is to transform visual observations into language descriptions that contain an actionable trait to it. SUM shares similar functionalities of visual captioning models, which aims to automatically generate fluent and informative language descriptions of an image (Ke et al., 2019). For the SUM to be capable of providing scene descriptions from visual observations, it needs to distill representative and meaningful visual representations from an image, then generate coherent and intelligent language descriptions.

In our framework, we adopt models with image captioning ability as our SUM. Generally, image captioning models employ a visual understanding system and a language model capable of generating meaningful and syntactically correct captions (Stefanini et al., 2021). In a standard configuration, the task can be defined as an image-to-sequence problem, where the inputs are pixels, which will be encoded as one or multiple feature vectors in the visual encoding step. Then a language model will take the information to produce a sequence of words or subwords decoded according to a given vocabulary in a generative way.

With the development of self-attention (Vaswani et al., 2017a), the visual features achieved remarkable performance due to multimodal pretraining and early-fusion strategies (Tan & Bansal, 2019; Lu et al., 2019; Li et al., 2020; Zhou et al., 2019). As for language models, the goal is to predict the probability of a given sequence of words occurring in a sentence. As such, it is a crucial component in image captioning, as it gives the ability to deal with natural language as a stochastic process. Formally, given a sequence of $n$ words $y_1, \ldots, y_n$, the language model compo-

nent of an image captioning algorithm assigns a probability $P(y_1, y_2, \ldots, y_n \mid \boldsymbol{X})$ to the sequence as:

$$P(y_1, y_2, \ldots y_n \mid \boldsymbol{X}) = \prod_{i=1}^{n} P(y_i \mid y_1, y_2, \ldots, y_{i-1}, \boldsymbol{X}) \quad (1)$$

where $\boldsymbol{X}$ represents the visual encoding on which the language model is specifically conditioned. Notably, when predicting the next word given the previous ones, the language model is autoregressive, which means that each predicted word is conditioned on the previous ones. Additionally, the language model usually decides when to stop generating captions by outputting a special end-of-sequence token.

### 3.3. APM: Decoding Language Information into Executable Action Plans

The goal of APM (action prediction module) is to transform latent language information from the SUM output into executable action plans. Since both latent language information and executable action plans are sequential data, a LLM with encoder-decoder architecture is a good option for APM in our framework. In addition, a LLM pretrained on a vast corpus of text already has adequate knowledge, which can be fine-tuned on other tasks to improve learning efficiency.

A LLM with encoder-decoder architecture suits well for our setting. The encoder is responsible for reading and understanding the input language information from SUM, which is usually based on transformer architecture, and creates a fixed-length vector representation, called the context vector. The decoder then takes the context vector as input and generates the output, in our case, the executable action plans. The decoder uses the context vector to guide its generation of the output and make sure it is coherent and consistent with the input information. However, due to the distribution change between the data that LLM was pretrained on and the new task, the LLM needs to be fine-tuned on the task-specific data to transfer the knowledge. The fine-tuning

**Algorithm 1** Fine-tuning SUM

Initialize pretrained SUM model
Load VirtualHome dataset for fine-tuning
**for** $n$ in num_epochs **do**
    **for** $\text{Image}_t$ and $\text{Caption}_t$ in $\text{batch}_n$ **do**
        1. $\hat{\text{Caption}}_t = \text{SUM}(\text{Image}_t)$
        2. $\text{Loss}_{XE_t}(\theta_t) = L_{XE}(\text{Caption}_t, \hat{\text{Caption}}_t)$
        3. $\theta_t \leftarrow \theta_t - \alpha\nabla_{\theta_t} L(\text{Caption}_t, \hat{\text{Caption}}_t)$
    **end for**
    **repeat**
        Steps 1 through 3
    **until** max(num_epochs) or convergence
**end for**

**Algorithm 2** Fine-tuning APM with Imitation Learning

Initialize fine-tuned SUM and pretrained APM
Load VirtualHome dataset for fine-tuning
**for** $n$ in num_epochs **do**
    **for** $\text{Image}_t$, $\text{Caption}_t$ $\text{Action}_t$ in $\text{batch}_n$ **do**
        1. $\hat{\text{Caption}}_t = \text{SUM}(\text{Image}_t)$
        2. $\hat{\text{Action}}_{t+1} = \text{APM}(\hat{\text{Caption}}_t, \text{Action}_t)$
        3. $\text{Loss}_{XE_t}(\theta_t) = L_{XE}(\text{Action}_t, \hat{\text{Action}}_{t+1})$
        4. $\theta_t \leftarrow \theta_t - \alpha\nabla_{\theta_t} L_{XE}(\text{Action}_t, \hat{\text{Action}}_{t+1})$
    **end for**
    **repeat**
        Steps 1 through 3
    **until** max(num_epochs) or convergence
**end for**

strategies will be introduced in the following sections. For our LLMs, we use well-adopted pretrained architectures, including BERT (Devlin et al., 2019), RoBERTa (Liu et al., 2019), and BART (Lewis et al., 2020), as both the encoder and decoder. The goal of the LLM is to learn how to generate programmable, executable actions from the language descriptions outputted by SUM.

### 3.4. Training Pipeline

The training pipeline contains two steps. We first fine-tune SUM with the curated VirtualHome observations (More details about data collection are introduced in Section 4.2). This fine-tuning step is to familiarize SUM with the types of scenes present in the task-specific data. We present pseudocode to fine-tune the SUM in Algorithm 1.

In the second stage, we load the fine-tuned SUM and encode the outputs as latent language embeddings. The embeddings are then fed into the APM, which is then fine-tuned using different fine-tuning loss objectives (supervised one or policy gradient, more details are introduced in Section 4), to achieve the optimal policy with maximum rewards. The pseudocode for finetuning APM with IL and REINFORCE are in Algorithms 2 and 3, respectively.

**Algorithm 3** Fine-tuning APM with REINFORCE

Initialize fine-tuned SUM, pretrained APM, and VirtualHome environment (env)
Load VirtualHome dataset for fine-tuning
**for** $n$ in num_epochs **do**
    $\text{Trajectories}_t = [\,]$
    $state = env.reset()$
    **for** $\text{Image}_t$, $\text{Caption}_t$ $\text{Action}$ in $\text{batch}_n$ **do**
        1. $\hat{\text{Caption}}_t = \text{SUM}(\text{Image}_t)$
        2. $\hat{\text{Action}}_t = \text{APM}(\hat{\text{Caption}}_t, \text{Action}_t)$
        3. $\text{Trajectories}_t.append(\hat{\text{Action}}_t)$
    **end for**
    $sort(\text{Trajectories}_t)$ by Task ID
    **for** $i$ in range(len($\text{Trajectories}_t$)) **do**
        4. $\hat{\text{Action}}_t = \text{sample\_action}(\text{Trajectories}_t[i])$
        5. $\text{Reward}_t = env.step(\text{Action}_t, \hat{\text{Action}}_t)$
        6. Compute $\nabla_{\theta_t} \log P(\hat{\text{Action}}_t|\text{Action}_t)$
        7. $\theta_t \leftarrow \theta_t + \alpha r\nabla_{\theta_t} \log P(\hat{\text{Action}}_t|\text{Action}_t)$
    **end for**
    **repeat**
        Steps 1 through 7
    **until** max(num_epochs) or convergence
**end for**

### 3.5. Fine-tuning APM with IL and RL

For LLM, the output word is sampled from a learned distribution over the vocabulary words. In the most simple scenario, i.e. the greedy decoding mechanism, the word with the highest probability is output. The main drawback of this setting is that possible prediction errors quickly accumulate along the way. To alleviate this drawback, one effective strategy is to use the beam search algorithm (Cho et al., 2014; Koehn, 2007) that, instead of outputting the word with maximum probability at each time step, maintaining $k$ sequence candidates and finally outputs the most probable one. For the training or fine-tuning strategies, most strategies are based on cross-entropy (CE) loss and masked language model (MLM). But recently, RL-based learning objective has also been explored, which allows optimizing for captioning-specific non-differentiable metrics directly.

**Imitation Learning with Cross-Entropy Loss** The CE loss aims to minimize the negative log-likelihood of the current word given the previous ground-truth words at each timestep. Given a sequence of target words $y_{1:T}$, the loss is formally defined as:

$$L_{XE}(\theta) = -\sum_{i=1}^{n} \log\left(P\left(y_i \mid y_{1:i-1}, \boldsymbol{X}\right)\right) \quad (2)$$

where $P$ is the probability distribution induced by LLM, $y_i$ the ground-truth word at time $i$, $y_{1:i-1}$ indicate the previous ground-truth words, and $\boldsymbol{X}$ the visual encoding. The cross-entropy loss is designed to operate at the word level and optimize the probability of each word in the ground-truth sequence without considering longer-range dependencies between generated words. The traditional training setting with

cross-entropy also suffer from the exposure bias problem (Ranzato et al., 2015) caused by the discrepancy between the training data distribution as opposed to the distribution of its own predicted words.

**Reinforcement Learning with REINFORCE**  Given the limitations of word-level training strategies observed when using limited amounts of data, a significant improvement was achieved by applying the RL approach. Under this setting, the LLM is considered as an agent whose parameters determine a policy. At each time step, the agent executes the policy to choose an action, i.e. the prediction of the next word in the generated sentence. Once the end-of-sequence is reached, the agent receives a reward, and the aim of the training is to optimize the agent parameters to maximize the expected reward (Stefanini et al., 2021).

Similar to Ranzato et al. (2015), for our policy gradient method, we use REINFORCE (Williams, 1992; Sutton et al., 1999), which uses the full trajectory, making it a Monte-Carlo method, to sample episodes to update the policy parameter. For fine-tuning LLMs using RL, we need to frame the problem into an Agent-Environment setting where the agent (policy) can interact with the environment to get the reward for its actions. This reward is then used as feedback to train the model. The mapping of the entities is from the agent (policy), which is an LLM, and the environment (the reward function, also named the model), which generates rewards. The reward function consumes the input as well as the output of the LLM to generate the reward. The reward is then used in a loss function, and the policy is updated. Formally, to compute the loss gradient, beam search and greedy decoding are leveraged as follows:

$$\nabla_\theta L(\theta) = -\frac{1}{k} \sum_{i=1}^{k} \left( \left( r\left(\boldsymbol{w}^i\right) - b \right) \nabla_\theta \log P\left(\boldsymbol{w}^i\right) \right) \quad (3)$$

where $\boldsymbol{w}^i$ is the $i$-th sentence in the beam or a sampled collection, $r(\cdot)$ is the reward function, and $b$ is the baseline, computed as the reward of the sentence obtained via greedy decoding (Rennie et al., 2016), or as the average reward of the beam candidates (Cornia et al., 2019). Note that, since it would be difficult for a random policy to improve in an acceptable amount of time, the usual procedure entails pretraining with cross-entropy or masked language model first, and then fine-tuning stage with RL by employing a sequence level metric as the reward. This ensures the initial RL policy is more suitable than the random one.

# 4. Experiments

This section introduces the environment we used in the experiments, the experimental settings, evaluations, and results. We would like to answer the following questions with experiments: (1) Can the proposed paradigm take pure

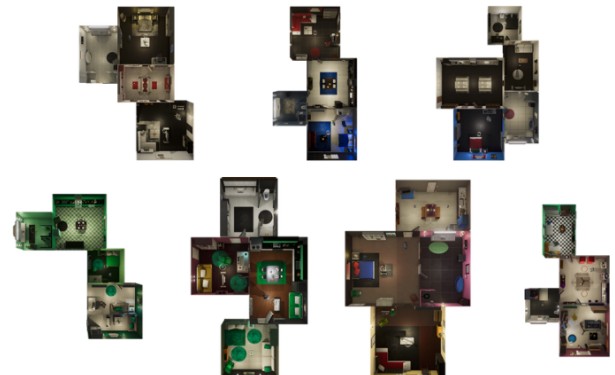

Figure 2. Top-down views of the seven different environments from VirtualHome (Puig et al., 2018b).

visual observations to generate executable robot actions; (2) What kinds of SUM are able to provide better scene descriptions for robot learning; (3) What kinds of APM show better action decoding ability in generating executable actions; (4) What kinds of fine-tuning strategies show better performance under this setting; (5) Can the model achieve consistent performance across different environments?

## 4.1. Environments and Metrics

**Environments**  We build the experiment environments based on VirtualHome (Puig et al., 2018a; Liao et al., 2019), a multi-agent, virtual platform for simulating daily household activities. (Puig et al., 2018b). Puig et al. (2018a) provides a dataset of possible tasks in their respective environments. Each task includes a natural language description of the task ("Put groceries in the fridge."), an elongated and more detailed natural language description of the task ("I put my groceries into the fridge."), and the executable actions to perform the task in the VirtualHome simulator ([[$Walk$] < $groceries$ > (1), [$Grab$] < $groceries$ > (1), ... [$Close$] < $fridge$ > (1)]). We define the training and testing tasks based on the natural language descriptions of the task due to their straightforwardness.

In VirtualHome, the agents are represented as 3D humanoid avatars that interact with given environments through provided, high-level instructions. Puig et al. (2018a) accumulated a knowledge base of instructions by using human annotators from AMT to first yield verbal descriptions of verbal activities. These descriptions were further translated by AMT annotators into programs utilizing a graphical programming language, thus amassing around 3,000 household activities in 50 different environments (Puig et al., 2018a). In this study, we evaluate our model's performance in seven unique environments in VirtualHome, which are shown in Figure 2. Each environment has a distinctive set of objects and actions that may be interacted with by agents.

**Metrics** We used standard NLP evaluation metrics, i.e., BLEU (Papineni et al., 2002), ROUGE (Lin, 2004), METEOR (Banerjee & Lavie, 2005), CIDEr (Vedantam et al., 2015), and SPICE (Anderson et al., 2016), for evaluating LLMs. In addition, we introduced the execution rate following Li et al. (2022b). The execution rate is defined as the probability of the agent's success in performing the outputted action from APM over the whole trajectory.

## 4.2. Datasets

To fine-tune SUM and APM on task-specific robot learning scenarios, we collect data via VirtualHome, including the agent's observations, language instructions, and action sequences. During data collection, a household activity program can be described as: $[[action_i < object_i > (id_i)], ... [action_n < object_n > (id_n)]]$, where $i$ denotes each step of the program, $action_i$ and $object_i$ denotes the action performed on the object at step $i$, and $id_i$ symbolizes the unique identifier of $object_i$ (Puig et al., 2018a). The original dataset was augmented by ResActGraph (Liao et al., 2019). After augmentation, the dataset contains over 30,000 executable programs, with each environment containing over 300 objects and 4,000 spatial relations. Additionally, we collect the image and text pairs separated by the environments they were executed in. This is important due to the different objects and actions available in each environment. However, as noted in Puig et al. (2018a) and Liao et al. (2019), not all programs were executable.

During data collection, we observed that the text was comprised of two words (e.g. walk bathroom, sitting chair, run treadmill). To have a more robust text description, we prompt engineered the texts with a fill-mask pipeline using BERT (Devlin et al., 2019; Song et al., 2019). For this study, we collect programs executed in three different views: 'AUTO', 'FIRST_PERSON', and 'FRONT_PERSON' as shown in Figure 3. In the 'AUTO' view, there are locked cameras in every scene through which the program randomly iterates through. The 'FIRST_PERSON' view observes the agent's actions through the first-person point of view. The 'FRONT_PERSON' view monitors the agent's actions through the front in a locked third-person point of view. Therefore, the final count of image-text pairs for our dataset in the 'AUTO', 'FIRST_PERSON', and 'FRONT_PERSON' views are 26,600, 26,607, and 26,608, respectively.

## 4.3. Experimental Setup

**SUM Setting** For SUM, we use the following image captioning models to serve as SUM: OFA (Wang et al., 2022), BLIP (Li et al., 2022a), and GRIT (Nguyen et al., 2022). Both OFA and BLIP are pretrained on the same five datasets, while the GRIT model (Nguyen et al., 2022) is pretrained on a different combination of datasets. For OFA, we adopted

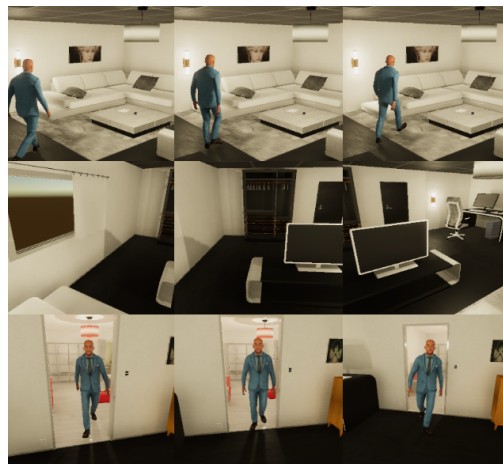

*Figure 3.* 'AUTO', 'FIRST_PERSON', 'FRONT_PERSON' views of the data collected from VirtualHome.

$OFA_{Large}$ due to its superior performance in five variations. $OFA_{Large}$ wields ResNet152 (He et al., 2015) modules with 472M parameters and 12 encoders and decoder layers. For BLIP, we used ViT-L/16 as the image encoder due to its better performance. For GRIP, we follow Nguyen et al. (2022) which utilizes the Deformable DETR (Zhu et al., 2020) framework. Note that in our study we want SUM to generate captions that not only describe the scene but also try to derive action from it. We observe that adding the prompt "a picture of " following Wang et al. (2021) causes the model to be biased in solely describing the scene, which would in turn not be helpful for generating actionable captions. Therefore, we remove prompts in the SUM setting. We load pretrained models and fine-tune them for 7 epochs on our collected VirtualHome dataset. We keep the hyperparameters consistent with the original implementations (Li et al., 2022a; Wang et al., 2022; Nguyen et al., 2022).

**APM Setting** We take LLM to act as the sole component in our APM. The goal of APM is to generate executable programs for the VirtualHome simulator. We deem the program outputted by the APM executable if the agent in the VirtualHome simulator is able to understand and perform the action. When the action is executed by the agent, the simulator is then directed to output images and captions that are synonymous with the input of SUM. The output hidden layers of SUM acts as the input embeddings to the APM, while the tokenized executable actions act as labels. The last hidden layer of APM acts as input embeddings for the tokenizer and generates token identifiers. The token identifiers are finally decoded into programmable actions that are fed into the VirtualHome simulator.

*Table 1.* Results by different SUM fine-tuned by imitation learning (IL) objective, where BERT serves as APM. The results are shown on 7 different environments in VirtualHome and also the average performance. The best result in each environment and each SUM model is marked in black and bold. The best SUM result with the highest average performance across 7 environments is marked in orange and bold.

| SUM/Results(%) | Environment | Bleu-1 | Bleu-2 | Bleu-3 | Bleu-4 | ROUGE-L | METEOR | CIDEr | SPICE | Execution Rate |
|---|---|---|---|---|---|---|---|---|---|---|
| OFA | 1 | 55.1±0.05 | 45.4±0.10 | 36.5±0.20 | 23.0±0.00 | 60.0±0.16 | 33.4±0.00 | 30.2±0.44 | **49.9**±0.43 | 78.0±2.39 |
| | 2 | 58.0±0.20 | 41.7±0.19 | 35.1±1.01 | 22.1±0.73 | 60.1±0.50 | 34.1±0.52 | 30.3±0.71 | 48.1±0.41 | 79.9±2.37 |
| | 3 | 55.3±0.30 | 42.3±0.62 | 34.9±0.15 | 23.0±0.00 | 60.5±0.01 | 34.8±0.64 | 31.2±0.55 | 48.4±0.17 | 80.0±3.29 |
| | 4 | 57.8±0.73 | 42.2±0.31 | 35.3±0.38 | 24.5±0.67 | 59.9±0.45 | 34.6±0.54 | 33.1±0.63 | 49.0±0.66 | 79.9±4.14 |
| | 5 | 59.4±0.44 | 40.3±0.03 | 34.8±0.02 | 24.2±0.37 | 59.7±0.25 | 35.1±0.62 | 32.7±0.24 | 38.0±0.13 | 77.4±1.12 |
| . | 6 | **60.5**±0.01 | **48.1**±0.53 | **36.6**±0.07 | **25.1**±0.15 | **61.9**±0.13 | **36.2**±0.60 | **34.6**±1.07 | **49.9**±0.77 | **80.5**±1.13 |
| | 7 | 58.2±0.30 | 46.5±0.58 | 34.6±0.04 | 22.3±0.08 | 58.3±0.92 | 35.6±0.62 | 30.8±0.37 | 44.2±0.33 | 69.2±2.31 |
| | Average | *57.8*±0.92 | **43.8**±1.02 | **35.4**±0.63 | **23.5**±0.77 | 60.1±0.41 | **34.8**±0.62 | **31.8**±1.31 | **46.8**±0.80 | **77.8**±3.26 |
| BLIP | 1 | 51.1±0.50 | 42.6±0.41 | 33.2±0.34 | 21.1±0.63 | 60.8±0.73 | 34.7±0.63 | **35.5**±00.09 | 42.7±0.91 | 72.6±1.99 |
| | 2 | 50.5±0.87 | 41.8±0.72 | 30.5±28 | 22.3±0.34 | 60.3±0.64 | 33.6±0.87 | 30.0±0.72 | 42.8±0.99 | 66.1±4.21 |
| | 3 | 52.4±0.54 | 43.2±0.65 | 33.6±0.13 | 21.1±0.52 | 61.4±0.29 | 34.5±0.12 | 31.1±0.00 | **48.9**±0.80 | 85.0±3.32 |
| | 4 | 51.0±1.19 | 42.1±0.87 | **33.8**±0.54 | 22.8±0.65 | 60.6±0.76 | 34.4±0.98 | 35.1±0.85 | 46.0±0.74 | 73.0±3.65 |
| | 5 | 49.0±0.53 | 38.8±0.43 | 30.4±0.72 | 20.0±0.47 | 58.6±0.65 | 34.1±0.75 | 21.0±0.66 | 30.8±0.69 | 67.2±0.93 |
| | 6 | **52.6**±0.79 | **44.5**±0.00 | 31.0±0.63 | **24.8**±0.62 | **62.0**±0.73 | **35.3**±1.02 | 31.0±0.02 | 42.4±0.87 | 84.1±3.54 |
| | 7 | 52.7±0.50 | 44.0±0.21 | 33.6±0.18 | 24.0±0.52 | 61.7±0.08 | 34.5±0.60 | 34.5±0.81 | 48.8±0.28 | **86.0**±4.92 |
| | Average | 51.3±0.31 | 42.4±0.54 | 32.3±0.66 | 22.3±0.31 | 60.7±0.63 | 34.4±0.75 | 31.2±0.87 | 43.2±0.97 | 76.3±5.22 |
| GRIT | 1 | 50.5±0.99 | 40.5±0.86 | 31.8±1.82 | 20.7±1.02 | 60.0±1.44 | 33.1±0.97 | 30.4±1.42 | 41.7±0.85 | 69.2±5.57 |
| | 2 | 52.1±0.66 | 41.8±1.77 | 31.7±1.92 | 20.1±0.97 | 59.9±0.65 | 32.1±0.76 | 29.4±0.87 | 42.0±0.88 | 71.4±5.52 |
| | 3 | 52.3±0.88 | 40.3±0.82 | 32.1±0.77 | 19.9±1.53 | 60.4±0.68 | 31.7±0.66 | 30.1±2.52 | 43.5±1.64 | 71.3±5.98 |
| | 4 | 51.9±0.93 | 39.8±0.92 | 31.8±0.97 | 21.3±1.72 | 59.7±1.22 | 32.0±0.76 | 30.0±0.79 | 42.8±0.84 | 72.8±4.65 |
| | 5 | **54.7**±0.93 | 42.3±1.02 | 33.2±1.25 | 24.5±0.93 | 62.3±1.42 | **33.8**±1.77 | 30.7±1.32 | **44.6**±1.23 | **78.5**±5.07 |
| | 6 | 54.6±1.42 | **44.7**±1.64 | **34.1**±1.32 | **25.8**±1.22 | **65.8**±1.25 | 30.1±2.31 | **34.5**±0.72 | 44.0±0.96 | 78.4±3.66 |
| | 7 | 53.9±0.88 | 42.0±1.79 | 32.6±2.00 | 22.5±0.90 | 63.4±1.00 | 31.8±1.23 | 32.3±1.31 | 43.1±1.41 | 70.0±3.99 |
| | Average | 52.9±0.18 | 41.6±0.87 | 32.4±0.72 | 22.1±0.68 | *61.6*±0.53 | 32.1±0.33 | 31.1±0.25 | 43.1±0.76 | 73.1±3.11 |

*Table 2.* Results by different APM fine-tuned by imitation learning (IL) loss objective. The results are shown by the average of 7 different environments in VirtualHome. The best results are marked in bold.

| APM/Results(%) | SUM | Bleu-1 | Bleu-2 | Bleu-3 | Bleu-4 | ROUGE-L | METEOR | CIDEr | SPICE | Execution Rate |
|---|---|---|---|---|---|---|---|---|---|---|
| BERT | OFA | **57.8**±0.92 | **43.8**±1.02 | **35.4**±0.63 | **23.5**±0.77 | 60.1±0.41 | **34.8**±0.62 | **31.8**±1.31 | **46.8**±0.80 | **77.8**±3.26 |
| | BLIP | 51.3±0.31 | 42.4±0.54 | 32.3±0.66 | 22.3±0.31 | 60.7±0.63 | 34.4±0.75 | 31.2±0.87 | 43.2±0.97 | 76.3±5.22 |
| | GRIT | 52.9±0.18 | 41.6±0.87 | 32.4±0.72 | 22.1±0.68 | **61.6**±0.53 | 32.1±0.33 | 31.1±0.25 | 43.1±0.76 | 73.1±3.11 |
| RoBERTa | OFA | **57.7**±0.01 | **43.2**±0.00 | **35.6**±0.48 | **24.1**±0.36 | 59.9±0.26 | **34.7**±0.51 | 31.4±0.47 | **47.3**±0.38 | 75.4±3.86 |
| | BLIP | 50.5±0.71 | 41.1±0.29 | 32.0±0.11 | 23.5±0.64 | **61.1**±0.88 | 33.0±0.70 | **31.8**±0.81 | 42.9±0.94 | **77.7**±0.71 |
| | GRIT | 53.1±1.02 | 42.0±0.90 | 34.1±1.01 | 23.1±1.22 | 60.4±1.92 | 31.5±0.59 | 31.5±1.42 | 42.8±1.77 | 75.4±4.39 |
| BART | OFA | **59.5**±0.09 | **45.9**±0.31 | **39.8**±0.37 | **28.1**±0.72 | 61.3±0.65 | **37.2**±0.69 | **34.4**±0.78 | 47.0±0.88 | **79.0**±1.91 |
| | BLIP | 52.9±0.80 | 44.3±0.52 | 35.5±0.49 | 25.3±0.62 | 62.2±1.12 | 35.3±1.62 | 32.0±0.97 | 44.5±0.88 | 76.0±1.98 |
| | GRIT | 54.2±1.68 | 43.2±1.85 | 33.6±1.60 | 25.3±0.93 | **62.7**±1.85 | 33.8±0.62 | 33.7±0.74 | 44.7±1.12 | 77.9±1.77 |

## 5. Results and Discussions

### 5.1. Model Performance with IL Fine-tuning

We first want to show the benefit of the proposed framework compared with other model architectures. Concretely, in the IL setting with expert data, we compare the execution rate of our model with the `MLP`, `MLP-1` and `LSTM` baselines in Li et al. (2022b). Our model has OFA in SUM and BART as APM. Note that all the baselines are not trained by datasets in other domains and have structured text input instead of realistic visual inputs as our proposed model. In the `LSTM` baseline, the hidden representation from the last timestep, together with the goal and current observation, are used to predict the next action. `MLP` and `MLP-1` both take the goal, histories, and the current observation as input and send them to MLPs to predict actions. `MLP-1` has three more average-pooling layers than `MLP` that average the features of tokens in the goal, history actions, and the current observation, respectively, before sending them to the MLP layer. More details about the baselines can be found in Li et al. (2022b).

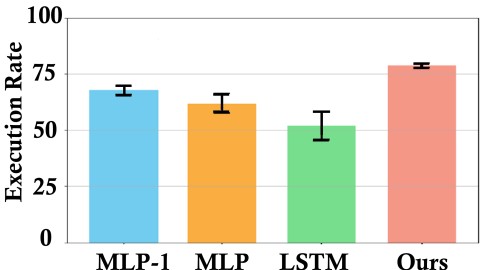

*Figure 4.* Comparison of our approach with baseline methods in the imitation learning setting evaluated by the execution rate.

As shown in Figure 4, our approach outperforms baselines in Li et al. (2022b) in terms of a higher average execution rate and a smaller standard deviation, though all the methods are trained on expert data with imitation learning objectives. The results show that the pretrained embeddings and large model architecture benefit the performance in downstream robot learning tasks.

*Table 3.* Execution Rates by different SUM fine-tuned by REINFORCE, where BERT serves as APM. The results are shown on 7 different environments in VirtualHome and also the average performance. The best results are marked in bold.

| SUM | Env-1 | Env-2 | Env-3 | Env-4 | Env-5 | Env-6 | Env-7 | Average |
|------|-----------|-----------|-----------|-----------|-----------|-----------|-----------|-----------|
| OFA | 50.1±0.65 | 50.3±0.52 | 51.5±0.48 | **57.8**±0.88 | 55.2±0.00 | 56.6±0.37 | **59.3**±0.48 | 54.4±0.55 |
| BLIP | **52.7**±0.78 | **53.4**±1.00 | **53.5**±0.92 | 55.6±0.68 | **60.1**±0.49 | **59.3**±0.91 | 49.9±0.90 | **54.9**±1.99 |
| GRIT | 38.7±1.02 | 40.0±1.11 | 51.3±0.99 | 48.2±0.90 | 46.5±0.85 | 55.8±0.70 | 45.3±1.08 | 46.5±2.01 |

*Table 4.* Results by different APM fine-tuned by REINFORCE loss objective. The results are shown by the average of 7 different environments in VirtualHome. The best results are marked in bold.

| APM | SUM | Execution Rate (%) |
|---------|------|--------------------|
| BERT | OFA | **54.7**±1.15 |
| | BLIP | 54.1±1.37 |
| | GRIT | 53.9±3.00 |
| RoBERTa | OFA | **55.6**±4.31 |
| | BLIP | 55.2±1.16 |
| | GRIT | 54.8±2.54 |
| BART | OFA | **57.2**±2.43 |
| | BLIP | 57.0±3.12 |
| | GRIT | 55.8±0.99 |

### 5.2. Model Performance with RL Fine-tuning

We provide the model performance after fine-tuning SUM with a frozen BERT in Table 1 for the IL setting with expert data and in Table 3 for the RL setting. We further provide the performance after fine-tuning APM with the fine-tuned SUM in Table 2 and Table 4. We can see that fine-tuning with expert data in IL results in higher average and per-environment performance than fine-tuning with RL, which shows the benefit of having access to the expert datasets. However, fine-tuning with RL still brings performance improvement to 57.2% as in Table 4. Note that without finetuning, the outputs of the LLMs in APM are generally not executable as shown in Figure 1. Moreover, we consistently observe that the combination of having OFA in SUM and BART as APM achieves the best performance after both IL (Table 2) and RL (Table 4) fine-tuning.

### 5.3. Ablation Study

To deeply understand the importance of different components in our paradigm that affect the overall performance, we conduct ablation studies on different factors including different components in SUM, different components in APM, and different environment variations.

**Different Components in SUM** The performances of using different components in SUM for IL and RL fine-tuning are in Table 1 and Table 3, respectively. From Table 1, we see that with expert data, OFA achieves better results than BLIP and GRIT on the average performance over 7 environments. We conjecture that this may be due to OFA being pretrained on 20M image-text pairs, which is larger than the size of other models' pretraining data. While under REINFORCE fine-tuning loss, as in Table 3, BLIP slightly

outperforms OFA in terms of average performance but has around 4 times larger standard deviation than OFA.

**Different Components in APM** The results of using different components in APM for IL and RL fine-tuning are presented in Table 2 and Table 4, respectively. We found that BART consistently outperforms other LLMs in both settings. We hypothesize that due to BART's architectural nature as a denoising autoencoder, it is more suitable for translating natural language descriptions into executable action programs for the VirtualHome simulator.

**Different Environments** To test the performance variations under different environments, we conducted the experiments separately for each unique environment. The results are shown in Table 1 and Table 3, for fine-tuning SUM under IL and RL settings, respectively. Due to image observation variations having the most impact on SUM instead of APM, so we only test the performance of SUM under different environment settings. Through Table 1 and Table 3, we could find that the variations exist among different environments. Generally, environment 6 seems to have the easiest environmental settings for the model to learn.

**Stability** To evaluate the stability of different models under different environments, we also calculated the standard deviation (stds) of the results across different trials. The results are shwon in Tables 1,2,3,4, which shows that BART as APM and OFA seem to be more stable than the rest of the combinations.

## 6. Conclusion

In this work, we introduce a novel robot learning paradigm with LLM in the loop that handles multiple modalities of visual observations and text-based actions in a principled manner. We bridge both modalities with natural language generated by a pretrained multimodal model. Our model contains SUM and APM, where SUM uses image observations as inputs taken by the robot to generate language descriptions of the current scene, and APM predicts the corresponding actions for the next step. We tested our method in the VirtualHome under 7 unique environments, and the results demonstrated that our proposed paradigm outperforms baselines in terms of execution rates and shows strong stability across environments. One interesting future direction is extending our proposed framework to solve generalization tasks in a more data and parameter-efficient manner.

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

## A. Experiment Parameters

*Table 5.* Experiment parameters used in SUMs, where the best ones are marked in bold

| SUM | Batch Size | Encoder Layers | Att. Heads | Learning Rate | Dropout | Epochs |
|-----|-----------|----------------|------------|---------------|---------|--------|
| OFA | [4, **8**, 16, 32] | [**24**] | [**16**] | [1e-4, **1e-5**, 1e-7] | [**0.1**, 0.2, 0.3] | [2, 5, **10**, 20, 50] |
| BLIP | [8, 16, **32**, 64] | [**12**] | [**12**] | [1e-4, **1e-5**, 1e-7] | [0.1, 0.2, **0.3**] | [2, **5**, 10, 20, 50] |
| GRIT | [4, 8, 16, **32**] | [**6**] | [**8**] | [**1e-4**, 1e-5, 1e-6] | [0.1, **0.2**, 0.3] | [2, 5, **10**, 20, 50] |

*Table 6.* Experiment parameters used in Supervised APMs, where the best ones are marked in bold

| APM | Batch Size | Encoder Layers | Att. Heads | Learning Rate | Dropout | Epochs |
|-----|-----------|----------------|------------|---------------|---------|--------|
| BERT | [4, **8**, 16, 32] | [**12**] | [**12**] | [1e-4, **1e-5**, 1e-7] | [0.1, 0.2, **0.3**] | [2, 5, **10**, 20, 50] |
| BART | [8, 16, **32**, 64] | [**12**] | [**16**] | [1e-4, **1e-5**, 1e-7] | [0.1, 0.2, **0.3**] | [2, 5, **10**, 20, 50] |
| RoBERTa | [4, 8, 16, **32**] | [**12**] | [**12**] | [**1e-4**, 1e-5, 1e-7] | [0.1, 0.2, **0.3**] | [2, 5, **10**, 20, 50] |

*Table 7.* Experiment parameters used in REINFORCE APMs, where the best ones are marked in bold

| APM | Batch Size | Encoder Layers | Att. Heads | Learning Rate | Dropout | Epochs |
|-----|-----------|----------------|------------|---------------|---------|--------|
| BERT | [4, **8**, 16, 32] | [**12**] | [**12**] | [1e-4, **1e-5**, 1e-7] | [0.1, 0.2, **0.3**] | [2, 5, **10**, 20, 50] |
| BART | [8, 16, **32**, 64] | [**12**] | [**16**] | [1e-4, **1e-5**, 1e-7] | [0.1, 0.2, **0.3**] | [2, 5, **10**, 20, 50] |
| RoBERTa | [4, 8, 16, **32**] | [**12**] | [**12**] | [1e-4, **1e-5**, 1e-7] | [0.1, **0.2**, 0.3] | [2, 5, **10**, 20, 50] |

## B. More Related Work

**Multimodal Learning**   Formalized multimodal learning research dates back to 1989 when (Yuhas et al., 1989) conducted an experiment that built off the McGurk Effect for audio-visual speech recognition using neural networks (Tiippana, 2014; McGurk & MacDonald, 1976). Researchers in NLP and CV collaborated to make large and multimodal datasets available, catering to specific downstream tasks, such as segmentation, detection, summarization, and so on (Xu et al., 2022; Qiu et al., 2022; Han et al., 2022; He et al., 2023; Qiu et al., 2023). In correlation, improvements in LLMs opened the gates to include other modalities of data, most frequently visual data (Wang et al., 2022; Nguyen et al., 2022; Li et al., 2022a; Wang et al., 2021; Shah et al., 2022; Zhang et al., 2021; Wang et al., 2020). By utilizing the learned embeddings that have been pretrained on both language and image datasets, vision-language models are able to perform very well. Within the above success, image captioning has been an important task in multimodal learning, which aims at generating textual descriptions for the given images. Recently, many models have been proposed and showed fabulous performances, i.e., BLIP, OFA, and GRIT (Li et al., 2022a; Wang et al., 2022; Nguyen et al., 2022).

## C. More Introduction about Different SUM

**OFA**   OFA (Wang et al., 2022) is a task and modality agnostic framework that supports a wide variety of cross-modal and unimodal tasks For the architecture, OFA utilizes a Seq2Seq (Sutskever et al., 2014) framework for all pretraining and downstream tasks of both cross-modal and unimodal generation. Data preprocessing and fixed modality accessories are necessary to warrant the joint training of visual and language data within the Transformer (Vaswani et al., 2017a) architecture. OFA uses ResNet (He et al., 2015) modules to convolve patch features of the hidden size during object feature extraction. In the case of processing language data, OFA follows the practice of GPT (Radford & Narasimhan, 2018) and BART (Lewis et al., 2019) by administering byte-pair encoding (BPE) for a given text sequence. The encoding is then transformed into a subword sequence, which is then embedded into features.

**BLIP**   BLIP (Li et al., 2022a) is a unified vision-language pretraining (VLP) framework that also supports a wide variety of vision-language tasks, such as image-text retrieval, image captioning, and visual question answering. The authors propose three functionalities in their multi-task model: unimodal encoder, image-grounded text encoder, and image-grounded text decoder (Li et al., 2022a). They adopt the visual transformer (ViT) (Dosovitskiy et al., 2020) as their image encoder. BLIP is able to effectively use noisy image and text pairs by bootstrapping the captions through their proposed method Caption and Filtering (CapFilt), in which a captioner, given web images, produces artificial captions and a filter removes the noisy image-text pairs.

**GRIT**   GRIT (Nguyen et al., 2022) is a Transformer-only (Vaswani et al., 2017a) architecture that uses grid and region-based features in images to generate captions. (Nguyen et al., 2022) define grid features as local image features extracted

from grid points and region features as a set of local image features of regions (i.e., bounding boxes). Instead of using a CNN-based object detector, GRID uses the Deformable DETR (Zhu et al., 2020) skeleton for faster computation (Nguyen et al., 2022). (Nguyen et al., 2022) also replaces the CNN backbone used in the original Deformable DETR to a Swin Transformer (Liu et al., 2021). The Swin Transformer (Liu et al., 2021) extracts features from the input image and obtains grid features as well. Similarly to the OFA model, the GRIT model was pre-trained with a cross-entropy loss and was fine-tuned using CIDEr-D optimization (Nguyen et al., 2022).

## D. More Experimental Results

**Fine-tuning performance on in-distribution tasks and unseen tasks** To further support our findings, we conducted additional experiments that tested the fine-tuning performance on in-distribution tasks and unseen tasks in the VirtualHome environment following the setting in Li et al. (2022b). Li et al. (2022b) used reinforcement learning to adapt to downstream tasks. It's important to note that Li et al. (2022b) used oracle text-based inputs that summarize the current observation, whereas we use raw image inputs and understand the scene with our fine-tuned SUM module. We measure the performance with the episode success rate and summarize the main comparison results with Li et al. (2022b)) in Table 8. Our results show that when fine-tuning with REINFORCE, our method outperforms Li et al. (2022b) in both in-distribution tasks and novel tasks. Additionally, when expert data is available in the downstream tasks, fine-tuning with imitation learning outperforms the REINFORCE approach.

*Table 8.* Comparison of episode success rate.

| Method | In-Distribution Tasks | Novel Tasks |
|---|---|---|
| Li et al. (2022b) | 53.7 | 27.8 |
| Ours (REINFORCE) | 58.4 | 33.7 |
| Ours (Imitation Learning) | 68.4 | 44.8 |

*Table 9.* Our fine-tuning results for different SUM/APM configurations in in-distribution and novel tasks, as well as using REINFORCE and imitation learning strategies. We measure the performance based on the episode success rate.

| SUM | APM | In-Distribution REINFORCE | Novel Tasks REINFORCE | In-Distribution Imitation | Novel Tasks Imitation |
|---|---|---|---|---|---|
| | BERT | 56.1 | 31.4 | 65.2 | 40.7 |
| OFA | BART | **58.4** | **33.7** | **68.4** | **44.8** |
| | RoBERTa | 51.7 | 32.3 | 66.0 | 42.8 |
| | BERT | 53.7 | 28.5 | 61.1 | 39.5 |
| BLIP | BART | 55.2 | 31.2 | 64.3 | 40.3 |
| | RoBERTa | 50.6 | 29.3 | 62.8 | 39.8 |
| | BERT | 50.5 | 28.8 | 61.3 | 40.4 |
| GRIT | BART | 51.2 | 30.0 | 63.7 | 39.6 |
| | RoBERTa | 49.0 | 27.1 | 59.2 | 38.7 |

**Importance and necessity of fine-tuning** To underscore the importance and necessity of fine-tuning, we present additional zero-shot testing performances without fine-tuning in Table 10 and Table 11. Our findings reveal that the episode success rate and action execution rates are significantly lower without fine-tuning in both methods, which highlights the crucial role that fine-tuning plays in improving performance.

*Table 10.* Comparison action execution rates in zero-shot and fine-tuned settings using both REINFORCE and Imitation Learning.

| Method | APM | SUM | REINFORCE | Imitation Learning |
|---|---|---|---|---|
| 1 | Zero-shot | Zero-shot | 0.1 | 0.1 |
| 2 | Zero-shot | Fine-tuned | 14.5 | 21.4 |
| 3 | Fine-tuned | Zero-shot | 5.8 | 6.9 |
| 4 | Fine-tuned | Fine-tuned | 57.2 | 77.8 |

*Table 11.* Comparison episode success rate in zero-shot and fine-tuned settings using both REINFORCE and Imitation Learning.

| Method | APM | SUM | REINFORCE | Imitation Learning |
|---|---|---|---|---|
| 1 | Zero-shot | Zero-shot | 0.7 | 0.7 |
| 2 | Zero-shot | Fine-tuned | 16.7 | 19.5 |
| 3 | Fine-tuned | Zero-shot | 7.7 | 8.7 |
| 4 | Fine-tuned | Fine-tuned | 58.4 | 76.8 |