# OpenReview forum: "Visual-based Policy Learning with Latent Language Encoding"
_ICML.cc/2023/Workshop/ILHF — ILHF Workshop ICML 2023_

### Official Review · Reviewer_hFGr · 2023-06-03
**Simple core approach and good results, but could improve clarity and framing.**

**Rating:** 7
**Confidence:** 4

**Review:**

Summary:
The authors propose a method for improving vision-conditioned policies with latent language. They outline a two-stage process wherein image observations are captioned with actionable text, which is then fed into a text-conditioned policy. All such steps are done via fine-tuned (visual) language models, with the action-conditioned part being trained via IL or RL on tasks in the VirtualHome environment. This system achieves higher execution rate than similar past works (e.g., that simply use a transformer model to integrate observations, goals, and histories, without any generative component).

Strengths:
- Proposed system achieves good results, in comparison to established past works.
- Extensive, well-presented results that highlight key points.
- Generally succeeds in addressing the research question: “[how to close] the… distribution shifts between the training tasks of large pretrained models and testing tasks in the domain of robot learning.”

Questions and Limitations:
- Could better motivate why using latent language captions for observations is useful in robotics (compared to e.g., training end-to-end or using a SayCan-like approach with a large VLM like PaLM-E). As it stands, it's unclear why this approach would be preferable to the many other ways of using (V)LMs as sources of prior knowledge for high-level semantic planning, especially when many of those seem to be zero-shot (while this one requires training and task-specific data).
  - SUM + APM seems to clearly outperform the LID equivalent (from Li et al., 2022b), but why exactly? What tasks did it improve on? Why did those tasks work with SUM + APM but not LID?
- Likewise, authors claim that a key challenge to “deploying LLMs to assist robots is that LLMs lack the capability to understand such non-text-based environment observations.” This could be better backed up: why do aforementioned VLMs not work well in terms of processing e.g., visual information for robotics?
- In keeping with the workshop theme, could frame above as improving interactive planning.
  - Generative captioning to produce latents for downstream policy-conditioning could be particularly useful for reasoning about what an observed person is doing and what actions should be taken in order to aid them. It could act as an approximation to goal inference (albeit not via a strict mathematical framework, like in Legibility and Predictability of Robot Motion by Dragan et al.), which allows the robot to attach goals, actions, and intents to observed entities, rather than just understanding the world in terms of pixels (or even objects)
- The title calls to mind Learning with Latent Language (Andreas et al., 2017). While that paper evaluates on a non-embodied task, the core idea – using language to describe inputs to “pick out” information for downstream tasks – is similar to the present work’s; would suggest citing it.
  - A key point in the above paper is that latent language helps with generalization (as it extracts information that can be useful in many tasks, akin to with meta learning). This work somewhat considers this fact, but notes that arbitrary captions are in general not very good for downstream robotics tasks (they’re not necessarily “actionable”).
- If there are 50 VirtualHome environments, why were only seven considered? That seems to be very few settings. Are there any ways to demonstrate that the system is generalizable and not just overfitting? How diverse are the task ranges? Any other related information (even train/test split sizes or strategies) would be appreciated.
- The algorithmic details and associated diagrams could be made a bit clearer. In particular:
  - The problem statement considers sequential observations producing sequential actions. However, the SUM formulation and its specific instantiations (e.g., BLIP) take in a single image at a time. It could be made more clear that this is happening
  - Likewise: The description for the APM is a bit unclear too. Is it just generating the next action, or a whole sequence? Is it conditioned on the original image (or its latent caption), or the latest one?
  - Figure 1 does show how the LMs may yield irrelevant outputs, however: (i) it’s not immediately clear why the fine-tuned SUM outputs are “actionable”: if “the woman … is walking to the kitchen,” what is the agent supposed to do about that?
  - Similarly, sure the fine-tuned APM outputs are executable, but how did that example action sequence (of cooking-related actions) arise from the observation captions?
- Additional/better illustrative examples could fix many of the above issues. It would be particularly useful to give actual captions and associated action sequences that the fine-tuned SUM + APM generated, as well as examples of RL vs IL APM action sequences.
- If space is needed for the illustrative examples (i.e., if displayed in figures), could move the per-environment results from Table 1 to the Appendix + just report averages. Especially true if more environments are introduced and the table takes more space.
- Very minor: There seems to be a floating citation in the first paragraph of 4.1 Environment and Metrics (Puig et al., 2018b). It is cited in a sentence with nothing else. Said citation also seems to be the same as Puig et al., 2018a (a is Arxiv, b is CVPR).

---

### Official Review · Reviewer_UV5e · 2023-06-06
**Review of Visual-based Policy Learning with Latent Language Encoding**

**Rating:** 9
**Confidence:** 4

**Review:**

The authors propose a novel LLM-based approach to robot learning that uses two modules (SUM and APM) to produce text summaries of an environment and then generate executable actions based on those summaries.

Strengths:
- The results clearly support the claims
- The exhaustive evaluation, including two learning methods (imitation learning and RL) and an ablation study, makes the results compelling
- The paper is clear and well-written

Minor comments:
- There are some minor typos and grammatical errors throughout and addressing these would further improve clarity.

---

### Decision · Program_Chairs · 2023-06-20

Accept